# Flying high on low cost: Success in the low-cost airline industry

**Veronika Majerová, Michal Jirásek** *

Faculty of Economics and Administration, Department of Business Management, Masaryk University, Brno, Czech Republic

* mijirasek@mail.municz

**Data Availability Statement:** All relevant data are within the paper and its Supporting information files.

**Funding:** The research was supported by the Masaryk University research project MUNI/A/1233/2022 Organizations in the era of uncertainty.

## Abstract

Low-cost airlines have embraced diverse business models, yielding varying degrees of success. In our study, we apply a configurational approach that allows us to evaluate business models not as isolated components but as intricate business configurations. Through this lens, we identify two distinct models that successful low-cost airlines adopt: the pure low-cost model and the hybrid model. Each model has its own specific, often contradictory, attributes. Most significantly, our findings indicate that low-cost airlines must choose between offering a broad spectrum of additional services or focusing on high productivity and on-time performance. Our analyses reveal that low-cost airlines cannot sidestep this trade-off, as a simultaneous offering of both models does not lead to success.

## Introduction

Navigating the turbulent skies of the air transportation industry places the business models of airlines under constant pressure [1]. Therefore, scholars have devoted considerable attention to identifying the critical factors in their success [2,3]. However, such studies often fall into the trap of analyzing success factors in isolation, overlooking the fact that these attributes are interrelated components of more comprehensive business models [4,5]. This underscores the need for a holistic evaluation that takes into account the synergistic effects and relational aspects of the business model attributes–an area in which conventional correlational methods fall short. A more appropriate tool for this task is Qualitative Comparative Analysis (QCA, [6,7]), which allows for a nuanced exploration of the complex interactions within business models ([8]–and illustrated by, e.g., [9]).

Our research focuses on a specific category of airlines—low-cost carriers (LCCs). Characterized by their streamlined, all-economy configurations and their use of secondary airports, these airlines offer a narrowed down service at reduced cost [10]. Despite their potential to disrupt the air transport market [11], LCCs face their own competitive pressures, prompting the ongoing evolution and diversification of their business models [12]. In this respect, determining what truly contributes to the success or failure of LCCs is paramount. Moreover, given the recent trend of traditional full-service carriers (FSCs) launching low-cost subsidiaries [13], understanding the conditions for success has become increasingly important.

**Competing interests:** The authors have declared that no competing interests exist.

This study aims to contribute to this understanding by highlighting the distinguishing features and limitations of successful LCC business models. While there have been many previous studies on airline (and LCC) business models (see below), most have relied on traditional correlation methods that assess the business model attributes in isolation. Although one study [9] has used configurational methods to look at other aspects of business models (i.e., configurations of different innovative activities), it differs from our research, which is focused on general attributes of LCC business models. Furthermore, by identifying distinct business models–the pure low-cost and hybrid models–and their specific attributes, our research provides invaluable insight into strategic decision-making in existing and emerging LCCs. Our findings thus advance the academic discourse on airlines' business models and offer practical implications for industry stakeholders seeking to navigate the complexities of the low-cost market.

## Theoretical overview

### Airline business models

As in other industries, distinctive business models constitute the basis of competition between individual airlines [10]. While some scholars have distinguished two [14] or more [10,15,16] categories of airline business models, others have argued for a more fluid understanding, suggesting that there is a spectrum of business models rather than simple categories [1].

Despite this ongoing debate, classifying business models still facilitates our understanding of airlines' operations. A standard classification [10,14] divides airlines into low-cost carriers (LCC) and full-service carriers (FSC). The business models of LCCs are characterized by numerous key attributes, including direct sales, significant outsourcing, high-density seating, high public awareness, and a focus on short-haul travel [10,17,18].

Moreover, FSCs and LCCs differ significantly in the design of their networks. FSCs often rely on a hub-and-spoke system, considered the most effective logistical system for moving passengers [19]. The hub is the airline's main base and is located in its country of origin. The airline operates direct lines from its hub to other destinations (i.e., the spokes). There is no direct connection between the spoke airports, and it is only possible to travel between them by transferring to another flight at the hub [20]. Thanks to the higher frequency of flights, this system offers economies of scale on connections and at hubs [21].

In contrast, LCCs typically operate on a point-to-point basis [10]. This system affords certain advantages, such as direct connections between airports, resulting in potential savings and convenience for passengers [14]. Nevertheless, it also has drawbacks, such as the low frequency of flights on routes, the need for a higher number of airplanes, low yields per seat, and the need for a high-density market to operate point-to-point flights [20].

In reality, distinguishing between business models is not black and white. As Lohmann and Koo [1] suggest, business models can be visualized on a continuum rather than as distinct entities. Today, many airlines adopt attributes from both FSC and LCC business models, creating a hybrid model that better caters to demand and competitive pressures [5,22].

A further convergence in business models can be observed in cases where FSCs establish or acquire low-cost subsidiaries [3,10,23]. However, these ventures often fail due to inappropriate role identification for the subsidiary LCC, overlaps in management, inadequate operational knowledge of the low-cost model, and other issues [13].

### Identification of business models

There are several approaches to identifying the business models of airlines [5]. For instance, Sengur and Sengur [24] base their conceptualization on multiple general business model frameworks, such as the Business Model Canvas. However, such broad applications may

present challenges in subsequent empirical studies. Daft and Albers' [25] approach is more tailored to empirical research and provides a comprehensive framework for describing the business models of airlines by means of three main components: corporate logic, value chain structure, and assets.

Yet it is the framework of Mason and Morrison [2] that seems to have attracted the most attention of researchers [5]. Their "Product and Organizational Architecture" (POA) offers a standardized method for categorizing key attributes, facilitating differentiation between airline models and their effects on profitability. Nevertheless, the POA framework also has shortcomings, one of which is that it assesses the relationship between components and profitability in isolation. Furthermore, subsequent studies have indicated difficulties in applying this framework due to data unavailability and the need for model adaptation (e.g., [1,26]).

In line with previous literature, we argue that business models are composites of various interrelated attributes that contribute to a complex structure that distinguishes each airline. A piecemeal assessment of the significance of these components (as in [2]) may mask their actual effects. Therefore, we view business models as exemplifying causal complexity and as defined by three features [27]: conjunction (the outcomes are not the product of a single cause but combinations of multiple conditions), equifinality (there can be more than one combination that leads to a particular outcome), and asymmetry (the conditions may have varying–or even contradictory–roles in different combinations).

One example of causal complexity is network types. There are successful airlines that operate either hub-and-spoke or point-to-point networks (e.g., Lufthansa and Ryanair), while others using the same network type do not fare as well (e.g., Eurowings and Thomas Cook). Although the network is crucial for success, it works best in conjunction with other conditions. There is also equifinality, as airlines with different network types can both succeed. However, a specific network type that is beneficial to one business model may not work for another, which is an illustration of asymmetry (e.g., an LCC using a hub-and-spoke network while maintaining low prices would not be sustainable due to the added transfer costs).

## Methodology

Our study concentrated on European airlines that followed various versions of the low-cost business model. We derived our sample of low-cost European airlines from a list provided by ICAO [28], subsequently analyzing 21 airlines that met our criteria and had adequate data for 2019. Our primary interest was to identify the business model attributes (conditions) that, when combined, either facilitated the airlines' success or hindered it.

To scrutinize the complex causality inherent in our case, we deployed a crisp-set Qualitative Comparative Analysis (QCA, e.g., [6]). Unlike conventional correlational approaches that examine conditions individually, QCA evaluates conditions collectively, highlighting their synergistic effects. This configurational approach embraces the asymmetry of conditions, indicating that the causes for an outcome do not necessarily mirror the causes for the absence of the said outcome [29]. Therefore, separate analyses need to be done for combinations leading to success and combinations leading to the absence of success among airlines.

A key feature of QCA is that it offers valid results with a moderate number of cases (10–100). Relative to qualitative methods, QCA accommodates more cases while preserving benefits such as in-depth case knowledge. Moreover, its variable-oriented approach promotes the identification of generalized variable relationships [6]. We used FsQCA 3.0 software for all the calculations.

## Qualitative comparative analysis in brief

QCA operates on principles based on Boolean algebra and Mill's methods [7]. Firstly, the selection of conditions (independent variables), the outcome (dependent variable), and the cases must be grounded in theoretical understanding. The data used in the analysis may be either qualitative or quantitative, but it must be convertible into binary codes (where 1 signifies the presence of a condition and 0 its absence). This is relevant for crisp-set QCA, which is the form of QCA we employed in our research.

QCA analysis involves multiple stages [6]. A critical step in the process is the creation of a "data matrix" and, subsequently, a "truth table." The data matrix is a simplified summary of binary values for each case, from which the truth table is derived. The truth table encompasses all empirically observed combinations of causal conditions. The following step involves the logical minimization of all combinations leading to the presence of the outcome (and, separately, combinations leading to the absence of the outcome). Logical minimization refers to the simplification of complex expressions into "prime implicants" according to Boolean algebra rules, which form the basis for interpreting the results. The findings are presented in three solutions–parsimonious, intermediate, and complex–which differ in the degree of simplification they offer.

## Business model attributes

Our study concentrates on the business models of LCCs and their connection to business success. We formulated six conditions (attributes of the business model) that differentiate various business models and might contribute to success or the absence of it: (i) operational size, (ii) membership in an airline group, (iii) breadth of services, (iv) focus on on-time service; (v) productivity, and (vi) provision of long-haul flights. The choice of these conditions was informed by the POA framework [2], other discussed business model frameworks, and a deep understanding of the individual cases. We focused on conditions that differentiate between the various LCC business models and excluded features which are commonly found in LCC models, such as paid onboard refreshments, the use of secondary airports, tight seat pitches, and limited seat widths. These conditions are common to all LCCs and, therefore, not decisive in distinguishing between successful and not successful ventures. We also did not include conditions (or their indicators) for which there are no openly available data for a sufficient number of airlines or where this data is largely incomplete (e.g., data related to the competition an airline is facing).

It should also be noted that we chose to work with these attributes regardless of whether they were found to be correlated to LCC success or its absence in previous studies. As we argued in the theoretical overview, findings from studies using correlational methods would be misleading, since they do not assess business model attributes as configurations (i.e., they do not assess how they work together and instead focus on their effects "in isolation").

We chose operational size (i) and airline group membership (ii) due to their potentially significant impact on business models, as they heavily influence management decisions and capital structure. Breadth of services (iii), on-time performance (iv), and productivity (v) are integral to the POA analysis and they are directly adjustable by airlines and leave room for innovation ([19] also supporting their critical role in LCC business models). We added (vi) the provision of long-haul flights as an additional feature that can distinguish between LCCs.

Table 1 displays the indicators for the individual conditions. The third column presents the criteria used for calibration, with the level of each quantitative condition derived from industry data (values that clearly divide the airlines into two subsets) or from the authors' knowledge of the industry and individual cases. We used the first approach (values that divide the airlines)

**Table 1. Conditions for QCA.**

| Condition | Indicator | Calibration criteria |
|---|---|---|
| Size (is large) | Availability of reports | Publishes own annual reports |
| | Number of destinations | More than 100 |
| | Number of employees | More than 3,000 |
| | Number of passengers | More than 15,000,000 |
| | Available seat kilometers | More than 30,000,000,000 |
| | Fleet size | More than 80 aircraft |
| Group (is a member of an airline group) | - | Another company has a controlling interest |
| Services (provides a broad breadth of services) | Through-ticketing | Offers through-ticketing |
| | Frequent flyer program | Offers a frequent flyer program |
| | Business class quality | Offers more than five features from: checked luggage, seat reservation, larger seat, warm meal, lounge access, priority check-in and boarding, change of bookings |
| On-time (focus on on-time service) | Proportion of on-time flights | Annual on-time performance is higher than 80% (median performance) |
| Productivity (has high productivity) | Fleet uniformity | One aircraft type constitutes more than 80% of the fleet |
| | Maximum seat capacity | The airline utilizes maximum seat capacity |
| | Available seat kilometers per employee | More than 10,000,000 |
| | Passengers per employee | More than 5,000 |
| | Employees per aircraft | Less than 50 |
| Long-haul (provides long-haul flights) | - | Offers long-haul flights (flights to South America, North America, Africa–excluding northern countries–or Asia–excluding the Middle East) |

where there is no (however abstract) threshold indicating the presence or absence of a condition. The best example for this case is the Size condition that works with airline-specific indicators. With six indicators, all airlines, apart from one, satisfy either a maximum of one criterion or five or six of them (the only exception, Jet2, satisfies two criteria). In most other cases, we combined our knowledge of where the threshold might be with insight into the actual criterion values (case and industry data).

Where appropriate, we employed several indicators to ensure the condition's validity and to decrease dependence on individual calibration criteria. Where there were multiple indicators, we marked the condition as present when any majority of the criteria of corresponding indicators was satisfied. We obtained the data for the analysis from various sources, including individual airlines' websites, other published documents and from industry sources. When data for any indicator was missing, we worked with the remaining indicators, the majority of which decided the presence or absence of a given condition.

Breadth of services includes the offer of through-ticketing (guaranteed compensation for missed connecting flights), a frequent flyer program, and business class quality. Although typically associated with FSCs, some LCCs have recently adopted these attributes. The focus on punctual service can be directly measured by OAG [30] statistics, with on-time flights defined as those that depart or arrive within 15 minutes of the scheduled time. High on-time performance enhances customer satisfaction while simultaneously improving the utilization of ground staff, aircrews, and aircraft.

**Table 2. Outcome for QCA.**

| Outcome | Indicator | Calibration criteria |
|---|---|---|
| Success | Load factor | More than 90% |
| | Cost per available seat kilometer | Less than 4 U. S. cents |
| | Evaluation of the parent company (only when it is a subsidiary) | Positive feedback from the parent company in annual reports |
| | Customer reviews | More than 3.5 points on Tripadvisor.com |
| | Growth in the number of passengers | The number of passengers grew between 2018 and 2019 |
| | Growth in the number of employees | The number of employees grew between 2018 and 2019 |
| | Overall growth (fleet, routes, destinations) | The number of airplanes, routes, and destinations grew between 2018–2019 |

We measured fleet and labor productivity by means of five indicators. The fleet is considered uniform if a single type of aircraft comprises over 80% of the fleet, which has a significant impact on operating costs and productivity. The second attribute is maximum seat capacity, which reflects the extent to which an airline utilizes its predominant aircraft type's maximum seat capacity. For example, the Boeing 737–800 has a maximum certified configuration of 189 seats [31]. A fleet operating at maximum seat capacity typically provides a single travel class and can sell more tickets per flight. Available seat kilometers per employee and passengers per employee are used interchangeably, given their high correlation in POA analysis [2].

Table 2 displays the indicators and corresponding criteria for the outcome. We identified seven indicators signifying success. However, we did not include any financial indicators due to the unavailability of data. For many of the observed airlines, financial reports are consolidated and fail to offer detailed information about subsidiaries. Thus, we chose alternative indicators with better availability of data. As before, we set the calibration criteria based on industry data.

## Results

In QCA analysis, reporting the inputs (the data matrix and the truth table) and outputs (analyses) in a research paper is standard. In the following section, we adopt QCA notation to streamline the textual presentation. More precisely, we use the symbol "~" to signify the absence of a condition or outcome (for example, an airline not offering long-haul flights is denoted as "~Long-haul") and the symbol "*" to indicate a combination of conditions (e.g., "~Long-haul*Group" signifies an airline that does not offer long-haul flights and which is a subsidiary of another airline).

### Data matrix and truth table

Table 3 represents the data matrix that serves as the starting point for QCA. It concisely summarizes the condition and outcome values for the 21 airlines examined. In the table "1" indicates the presence of a condition or outcome, while a "0" signifies its absence (marked as "~" in the text).

Table 4 presents the truth table that includes the 14 combinations of causal conditions and outcomes observed in our data. Given that there are six conditions, there are 64 ($2^6$) theoretically possible combinations, for which it is clearly impossible for 21 airlines to represent. However, as Ragin [32] points out, limited diversity is natural, as the empirical world seldom depicts all logically conceivable combinations.

**Table 3. Data matrix.**

| Airline | Size | Group | Services | On-time | Productivity | Long-haul | Success |
|---|---|---|---|---|---|---|---|
| Condor | 0 | 1 | 1 | 0 | 0 | 1 | 1 |
| Eurowings | 1 | 1 | 1 | 1 | 0 | 0 | 0 |
| Iberia Express | 0 | 1 | 1 | 1 | 0 | 0 | 0 |
| Laudamotion | 0 | 1 | 0 | 0 | 1 | 0 | 0 |
| SunExpress | 0 | 1 | 1 | 1 | 0 | 0 | 0 |
| Transavia | 1 | 1 | 0 | 0 | 1 | 0 | 1 |
| Level | 0 | 1 | 1 | 1 | 0 | 1 | 0 |
| Vueling | 1 | 1 | 0 | 0 | 1 | 0 | 1 |
| easyJet | 1 | 0 | 1 | 0 | 0 | 0 | 1 |
| Jet2 | 0 | 0 | 0 | 1 | 1 | 0 | 1 |
| Norwegian | 1 | 0 | 1 | 0 | 0 | 1 | 1 |
| Ryanair | 1 | 0 | 0 | 1 | 1 | 0 | 1 |
| SmartWings | 0 | 0 | 0 | 0 | 0 | 0 | 0 |
| Volotea | 0 | 0 | 0 | 1 | 1 | 0 | 1 |
| Wizz Air | 1 | 0 | 0 | 1 | 1 | 0 | 1 |
| Albawings | 0 | 0 | 0 | 1 | 1 | 0 | 1 |
| Blue Air | 0 | 0 | 1 | 0 | 0 | 0 | 1 |
| Blue Panorama | 0 | 0 | 0 | 0 | 1 | 1 | 0 |
| Helvetic Airways | 0 | 0 | 1 | 0 | 0 | 0 | 1 |
| Pegasus Airlines | 1 | 0 | 1 | 0 | 0 | 0 | 1 |
| Pobeda | 0 | 1 | 0 | 1 | 1 | 0 | 1 |

## Analysis of necessary conditions

In addition to identifying combinations of conditions sufficient for the outcome, QCA also allows for the analysis of the necessary conditions. It is important to understand that the necessary conditions can be sufficient or insufficient for the outcome. The results for both success and its absence are displayed in Table 5, which are based on two separate analyses. Consistency denotes the ratio of cases that have both the condition and the outcome. According to Ragin

**Table 4. Truth table.**

| Size | Group | Services | On-time | Productivity | Long-haul | Success | Number of cases |
|---|---|---|---|---|---|---|---|
| 0 | 0 | 0 | 1 | 1 | 0 | 1 | 3 |
| 0 | 0 | 1 | 0 | 0 | 0 | 1 | 2 |
| 0 | 1 | 0 | 1 | 1 | 0 | 1 | 1 |
| 0 | 1 | 1 | 0 | 0 | 1 | 1 | 1 |
| 0 | 1 | 1 | 1 | 0 | 0 | 0 | 2 |
| 0 | 0 | 0 | 0 | 0 | 0 | 0 | 1 |
| 0 | 1 | 0 | 0 | 1 | 0 | 0 | 1 |
| 0 | 1 | 1 | 1 | 0 | 1 | 0 | 1 |
| 0 | 0 | 0 | 0 | 1 | 1 | 0 | 1 |
| 1 | 0 | 1 | 0 | 0 | 0 | 1 | 2 |
| 1 | 1 | 0 | 0 | 1 | 0 | 1 | 2 |
| 1 | 0 | 0 | 1 | 1 | 0 | 1 | 2 |
| 1 | 0 | 1 | 0 | 0 | 1 | 1 | 1 |
| 1 | 1 | 1 | 1 | 0 | 0 | 0 | 1 |

**Table 5. Analysis of necessary conditions.**

| Outcome variable: Success | | | Outcome variable: ~Success | | |
| --- | --- | --- | --- | --- | --- |
| Conditions tested | Consistency | Coverage | Conditions tested | Consistency | Coverage |
| Size | 0.500 | 0.875 | Size | 0.143 | 0.125 |
| Group | 0.286 | 0.444 | Group | 0.714 | 0.556 |
| Services | 0.429 | 0.600 | Services | 0.571 | 0.400 |
| On-time | 0.429 | 0.600 | On-time | 0.571 | 0.400 |
| Productivity | 0.571 | 0.800 | Productivity | 0.286 | 0.200 |
| Long-haul | 0.143 | 0.500 | Long-haul | 0.286 | 0.500 |
| ~Size | 0.500 | 0.538 | ~Size | 0.857 | 0.462 |
| ~Group | 0.714 | 0.833 | ~Group | 0.286 | 0.167 |
| ~Services | 0.571 | 0.727 | ~Services | 0.429 | 0.273 |
| ~On-time | 0.571 | 0.727 | ~On-time | 0.429 | 0.273 |
| ~Productivity | 0.429 | 0.545 | ~Productivity | 0.714 | 0.455 |
| ~Long-haul | 0.857 | 0.706 | ~Long-haul | 0.714 | 0.294 |

[32], a consistency of 0.9 indicates a necessary condition. Coverage, on the other hand, reveals the empirical relevance of a condition in the sample, computed as the ratio of cases possessing the condition in the total number of cases. None of the conditions in this analysis can be deemed necessary, as they fail to pass the consistency threshold of 0.9. Only the consistencies of the ~Long-haul (outcome: Success) and ~Size (outcome: ~Success) conditions come close to this figure, both of which recorded consistencies of 0.857.

## Analysis of sufficient conditions

To simplify the QCA output, we will now concentrate on the intermediate solutions, which are typically those most frequently interpreted [6]. The intermediate solutions for the presence and absence of success are presented in Tables 6 and 7, while the complex and parsimonious solutions can be found in the supporting information section (S1–S4 Tables). Each table is divided into four columns: the first outlines the combinations of conditions determined by the analysis; the second presents the raw coverage (the ratio of cases with the outcome possessing this combination); the third presents the unique coverage (the ratio of cases with the outcome possessing this combination and none of the others); and the fourth pertains to consistency (the proportion of cases with a combination that also exhibits the outcome).

## Intermediate solutions (Success)

Some unexpected findings were revealed regarding the combinations of conditions linked to success (Table 6). Despite the potential correlational-logic supposition that the On-time condition would be present in the Success outcomes, we identified three combinations where this condition was absent. However, these only accounted for 29% of the Success outcome. By contrast, a single solution featuring the On-time condition covered 43% of the Success outcome. As we will discuss later, this potentially indicates that LCCs face a trade-off. Specifically, maintaining on-time flights appears incompatible with offering a broad range of services (the presence of the Services condition).

Notably, it seems that being a subsidiary does not play a significant role. The combinations that include the presence and absence of this condition (Group) appear equally in combinations leading to success, and the condition is absent in combination with the highest coverage (0.429, ~Services*On-time*Productivity*~Long-haul). Productivity is present in two

**Table 6. Intermediate solutions (Success).**

| | Raw coverage | Unique coverage | Consistency |
|---|---|---|---|
| ~Group*Services*~On-time*~Long-haul | 0.286 | 0.143 | 1 |
| Size*~Group*Services*~On-time | 0.214 | 0.071 | 1 |
| ~Services*On-time*Productivity*~Long-haul | 0.429 | 0.429 | 1 |
| Size*Group*~Services*Productivity*~Long-haul | 0.143 | 0.143 | 1 |
| ~Size*Group*Services*~On-time*Long-haul | 0.071 | 0.071 | 1 |
| Solution coverage: 1 | | | |
| Solution consistency: 1 | | | |
| Assumptions: On-time (present), Productivity (present) | | | |

combinations (~Services*On-time*Productivity*~Long-haul and Size*Group*~Services*Pro-ductivity*~Long-haul) covering 50% of cases. Its absence does not feature in any of the combinations. We are cautious when interpreting the role of long-haul flights, given that this condition only appears in four cases. Nevertheless, it seems that successful European LCC airlines tend not to have this condition (and therefore only operate within Europe and its neighboring regions).

## Intermediate solutions (~Success)

The combinations of conditions leading to the absence of success (Table 7) support the general trade-off between on-time performance and breadth of services. All of the airlines that were not successful exhibited either both or neither of these two conditions (see all six combinations of conditions in Table 7). The airlines that were not successful were typically small (~Size feature in five out of six combinations that are part of the intermediate solution for ~Success), although this may not apply to airlines that are subsidiaries (Group). It could be argued that for airlines that are failing, maintaining their size is challenging unless they have a parent company to provide financial support.

## Interpretation of results

Our analysis reveals two distinct business models that contribute to the success of LCCs (see Table 6). The first, which we term the "hybrid model", closely resembles an FSC in its broad range of services. However, this model tends to compromise when it comes to on-time performance (Services*~On-time). In our study, airlines such as easyJet and Norwegian exemplify this model.

**Table 7. Intermediate solution (~Success).**

| | Raw coverage | Unique coverage | Consistency |
|---|---|---|---|
| ~Size*Group*~Services*~On-time*~Long-haul | 0.143 | 0.143 | 1 |
| ~Size*~Group*~Services*~On-time*Long-haul | 0.143 | 0.143 | 1 |
| Group*Services*On-time*~Productivity*~Long-haul | 0.429 | 0.143 | 1 |
| ~Size*Group*Services*On-time*~Productivity | 0.429 | 0.143 | 1 |
| ~Size*~Services*~On-time*~Productivity*~Long-haul | 0.143 | 0 | 1 |
| ~Size*~Group*~Services*~On-time*~Productivity | 0.143 | 0 | 1 |
| Solution coverage: 1 | | | |
| Solution consistency: 1 | | | |
| Assumptions: ~On-time (absent), ~Productivity (absent) | | | |

The second successful business model, which we term the "pure low-cost model", is characterized by a limited range of services and the absence of long-haul flights (~Services*~Long-haul). A limited offer of services, such as catering, enabling rapid aircraft turnaround and improved on-time performance facilitates this business model's efficiency [33]. The model aligns with Mason and Morrison's [2] characterization of a successful LCC. In our sample, Ryanair and WizzAir are examples of this model.

While successful airlines cannot simultaneously offer a broad range of services and maintain a highly productive/on-time business model (see Table 7 and its interpretation above), they must adopt one of these approaches. Either they can copy aspects of the FSC business model or focus on delivering the core product with cost efficiencies. To illustrate the trade-offs airlines face, we have constructed a trade-off triangle, which is depicted in Fig 1.

According to the notional trade-off triangle, LCCs cannot afford the absence of productivity (~Productivity; see also the argument by [19]). As they compete on low prices, they need to maintain business efficiencies, particularly when they do not offer additional services (~Services), as is the case for the pure low-cost model. The hybrid model allows for some flexibility in this area but does so at the expense of on-time performance (~On-time) and, potentially, the core product itself. In both models, LCCs must appeal to customers by offering additional services (Services) or delivering high on-time performance (On-time). However, they cannot provide both without jeopardizing their overall success. Such an offering would essentially mean adopting a traditional FSC model while retaining low prices, which is an untenable

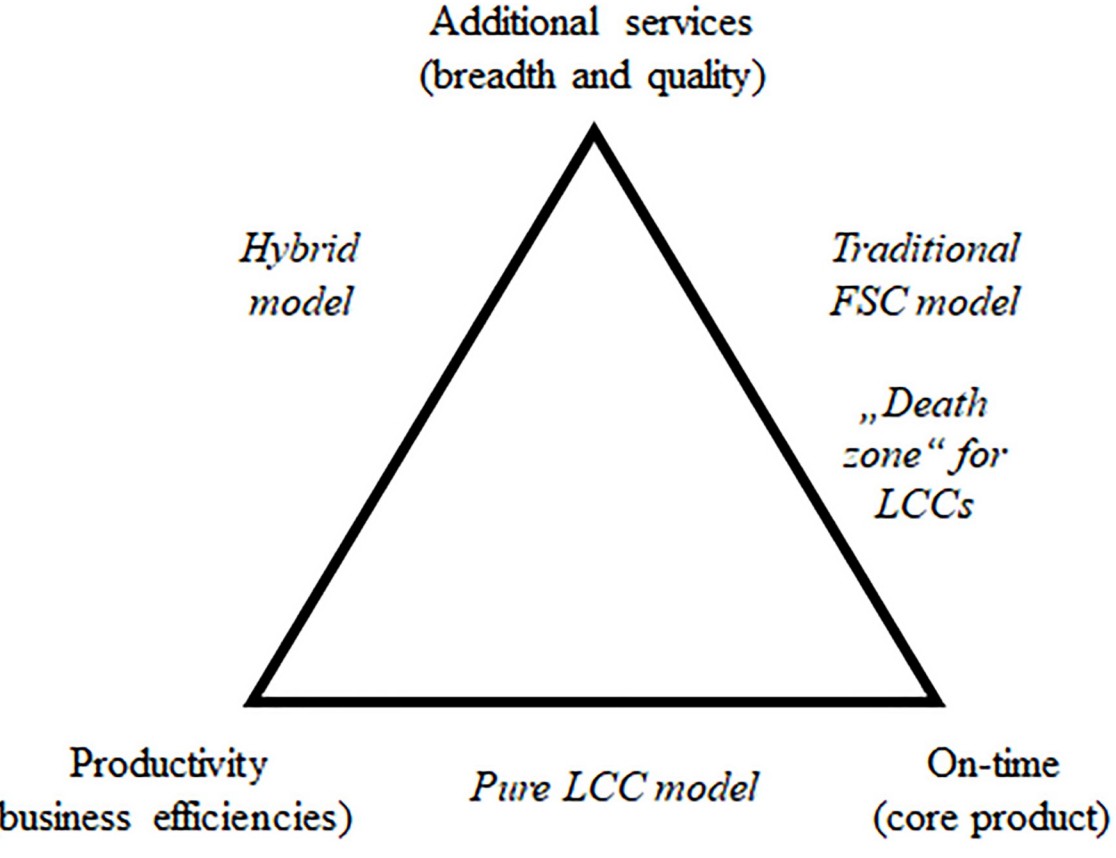

**Fig 1. Trade-offs in the business models.**

strategy in the long term. This is reflected in our analysis of LCCs that were not successful (Table 7).

Finally, being a subsidiary seems to be somewhat of a liability for LCCs (and as Gillen and Gados, [19], argue, most such ventures have failed). This observation is supported by the frequent exits of LCC subsidiaries from the market segment [34]. Overlapping management and unclear subsidiary roles may contribute to this issue [13].

## Discussion and conclusion

The configurational logic guiding our research enables us to evaluate LCC business models as complex combinations of diverse attributes [8]. We argue that this study reflects the three features of causal complexity [27]. Firstly, our results uphold the conjunction principle–success or failure cannot be traced back to a single attribute. Instead, it results from various business model attributes operating in conjunction. Secondly, we observed equifinality in our findings, having identified two prevalent approaches adopted by successful airlines: the pure low-cost and hybrid business models. Finally, we found asymmetry, which means that the individual attributes exerted different effects when combined with others. We view this last characteristic as the fundamental strength of our approach, contrasting with traditional correlational logic (as adopted in, for instance, [2]).

To illustrate the asymmetry, consider the attribute of a broad range of services. This feature is common throughout our sample and appears with successful and unsuccessful airlines. Thus, a correlation or regression analysis would likely yield a non-significant or a slightly significant relationship. However, we demonstrated that a broad range of services is a critical element in one business model of successful LCCs: the hybrid model. While this attribute might be overlooked in a correlational analysis, it is crucial to one of the pathways to success.

Hence, we argue that our research sheds additional light not on the attributes of business models, which are widely acknowledged in the industry, but on the relationships between these attributes. The correlational approach calls for the adoption of individual best practices. However, our study suggests that the ill-considered adoption of industry practices could potentially harm an organization if these practices are incompatible with its current business model. This observation probably accounts for why some FSC low-cost subsidiaries have succeeded while others have exited the sector [34].

Our study employed crisp-set Qualitative Comparative Analysis (QCA), chosen for its straightforward interpretation and communication of results. Nevertheless, this approach is constrained by its binary value system, which acknowledges a condition as either present or absent. Fuzzy set QCA offers a more nuanced approach that accommodates the subtleties observed in real life [6]. This is a promising avenue for future research that may lead to more robust findings than one which relies on crisp-set QCA.

Our analysis concentrated on attributes that distinguish between alternative LCC business model features. Given that many attributes are incorporated into most LCC models (e.g., paid onboard refreshments, use of secondary airports, small seat pitches and seat width), we argue that they play a minimal role in differentiating between success and failure. Therefore, these attributes were excluded from our analysis, which allowed us to focus on differentiating LCC features. However, this might inadvertently suggest that these attributes are unimportant. While we recognize that some business model attributes might be redundant, many others must be present in every LCC business model. Identifying these conditions would be a suitable task for the related Necessary Condition Analysis method [35], in the case where it was supplemented with more detailed data on these attributes. Such an analysis could yield important insights for LCCs considering abandoning or limiting these practices.

Finally, our analysis was based on European LCCs. Given the differences in regional air transportation markets (reflected in, e.g., [18]), our study would require replication in other markets to extend its findings to them.

In conclusion, our research identified two distinct business models adopted by successful LCCs: the pure low-cost model and the hybrid model. Specifically, LCCs must choose between offering a broad range of additional services and focusing on their core product, represented by on-time performance and high productivity. While they must select one of these models to attract customers, they cannot adopt both without rendering their model unworkable. By employing a configurational approach, we examined the fundamental attributes of business models holistically, rather than treating them as separate factors. We believe this approach is particularly beneficial when investigating such complex phenomena.

## Supporting information

**S1 Table. Complex solution (success).**
(DOCX)

**S2 Table. Parsimonious solution (success).**
(DOCX)

**S3 Table. Complex solution (~success).**
(DOCX)

**S4 Table. Parsimonious solution (~success).**
(DOCX)

**S1 Data.**
(XLSX)

## Author Contributions

**Conceptualization:** Michal Jirásek.

**Data curation:** Veronika Majerová.

**Formal analysis:** Veronika Majerová.

**Funding acquisition:** Michal Jirásek.

**Methodology:** Veronika Majerová, Michal Jirásek.

**Project administration:** Veronika Majerová.

**Software:** Veronika Majerová.

**Supervision:** Michal Jirásek.

**Visualization:** Michal Jirásek.

**Writing – original draft:** Michal Jirásek.

**Writing – review & editing:** Michal Jirásek.

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
