## [Decision Letter · Decision Letter 0]

4 Apr 2023

PONE-D-22-33605What makes a low-cost airline successful? A configurational approachPLOS ONE

Dear Dr. Jirásek,

Thank you for submitting your manuscript to PLOS ONE. After careful consideration, we feel that it has merit but does not fully meet PLOS ONE’s publication criteria as it currently stands. Therefore, we invite you to submit a revised version of the manuscript that addresses the points raised during the review process.

I would like to sincerely apologise for the delay you have incurred with your submission. It has been exceptionally difficult to secure reviewers to evaluate your study. We have now received two completed reviews; the comments are available below. The reviewers have raised significant scientific concerns about the study that need to be addressed in a revision. Ensure that suggested additional references are in fact relevant - inclusion of references suggested by the referees is not a prerequisite for acceptance of your manuscript. 

Please revise the manuscript to address all the reviewer's comments in a point-by-point response in order to ensure it is meeting the journal's publication criteria. Please note that the revised manuscript will need to undergo further review, we thus cannot at this point anticipate the outcome of the evaluation process.

We look forward to receiving your revised manuscript.

Kind regards,

Miquel Vall-llosera Camps

Senior Editor

PLOS ONE

Journal Requirements:

Reviewers' comments:

Reviewer's Responses to Questions

**Comments to the Author**

1. Is the manuscript technically sound, and do the data support the conclusions?

Reviewer #1: Yes

Reviewer #2: Partly

2. Has the statistical analysis been performed appropriately and rigorously? 

Reviewer #1: Yes

Reviewer #2: I Don't Know

3. Have the authors made all data underlying the findings in their manuscript fully available?

Reviewer #1: No

Reviewer #2: No

4. Is the manuscript presented in an intelligible fashion and written in standard English?

Reviewer #1: No

Reviewer #2: No

5. Review Comments to the Author

Reviewer #1: Comments on Manuscript “What makes a low-cost airline successful? A configurationally approach” Thank you very much for giving me the opportunity to review the manuscript.

I must appreciate the work authors have performed.

Low-cost airlines have adopted a wide variety of business models that have met with varying levels of success. In our research, we use a configurationally approach that enabled us to assess business models not in terms of their separate components but as complex business configurations. In this way, we were able to identify two distinct models that successful low-cost airlines adopt: the pure low-cost model and the hybrid model. Each has its specific attributes, which often contradict those of the other. Most importantly, this implies that low-cost airlines need to choose between offering a broad range of additional services or focusing on high productivity and on-time performance. Our analyses show that low-cost airlines cannot escape this trade-off: They cannot offer both to be successful.

I have reviewed this paper thoroughly and a few suggestions are given below:

Introduction:

I would suggest improving this section by making it more explainable in terms of the Lower cost airlines with the significance of configurationally approach included in the study.

Avoid using abbreviations (explain it in detail in the first stance and then use the acronyms.)

Literature Review:

This section is well-aligned and presents an effective blend of recent and past studies. Overall, this section is appropriate in my opinion. Kindly add some studies in literature review section;

https://doi.org/10.1002/ijfe.2073

https://doi.org/10.1007/s11356-022-21929-w

https://doi.org/10.1177/0958305X221134113

https://doi.org/10.1007/s11356-022-22672-y

https://doi.org/10.3389/fenvs.2022.967418

10.3389/fpubh.2022.1009393

https://doi.org/10.3390/su14031054

https://doi.org/10.1007/s11356-021-17438-x

https://doi.org/10.1108/LHT-03-2021-0113

https://doi.org/10.1007/s11356-022-19718-6

https://doi.org/10.1007/s11356-022-19628-7

https://doi.org/10.1108/FS-02-2021-0053

https://doi.org/10.1016/j.resourpol.2022.102730

https://doi.org/10.1007/s11356-022-19954-w

https://doi.org/10.1007/s11356-022-20178-1

https://doi.org/10.1007/s11356-022-20922-7

Methodology:

This section is well-written and well-explained. Referring to my comments in the Introduction section, try to explain each abbreviation at first and then use its short form.

Empirical Results:

Results are explained in detail. Must add much more explanations and interpretations for the results, which are not enough. It is suggested to compare the results of the present research with some similar studies which is done before (more justification is needed).

Conclusion:

Please make sure your conclusions section underscores the scientific value-added of your paper and the applicability of your findings/results, as indicated previously. Please revise your conclusion part into more detail. It would be best if you enhanced your contributions, and limitations, underscore the scientific value-added of your paper, and the applicability of your findings/results and future study in this session.

I hope these comments would enhance the quality of the manuscript to make it an appropriate fit for the Journal of PLOS ONE Readership.

Reviewer #2: The study is interesting but requires few changes mentioned below

1. References used in the study aren't recent ones. Please add more of recent research on the topic.

2. The motivation to or need to conduct this study is missing.

3. Please consider correcting grammatical errors as in Line 202

4. Title of the study doesn't appear interesting, it should be changed.

6. PLOS authors have the option to publish the peer review history of their article (what does this mean?). If published, this will include your full peer review and any attached files.

Reviewer #1: **Yes: **KASHIF ABBASS

Reviewer #2: No

---

## [Author Response · Author response to Decision Letter 0]

9 Jun 2023

Dear editor,

We want to thank you and both reviewers for the feedback and the opportunity to review the paper. We substantially rewrote the whole manuscript and made numerous changes to make it better and more interesting for the audience. Below we provide a response to the reviewers’ comments:

Reviewer #1

Introduction: I would suggest improving this section by making it more explainable in terms of the Lower cost airlines with the significance of configurationally approach included in the study.

Avoid using abbreviations (explain it in detail in the first stance and then use the acronyms.)

We changed the introduction to clarify why the study is interesting and why it is helpful to use the configurational approach. We separated this argument from the rationale of studying low-cost air carriers – we believe the changes make the introduction much more welcoming and straightforward.

We limited the number of abbreviations we used (not only in the introduction but also in the rest of the paper).

This section is well-aligned and presents an effective blend of recent and past studies. Overall, this section is appropriate in my opinion. Kindly add some studies in literature review section;

https://doi.org/10.1002/ijfe.2073

https://doi.org/10.1007/s11356-022-21929-w

https://doi.org/10.1177/0958305X221134113

https://doi.org/10.1007/s11356-022-22672-y

https://doi.org/10.3389/fenvs.2022.967418

10.3389/fpubh.2022.1009393

https://doi.org/10.3390/su14031054

https://doi.org/10.1007/s11356-021-17438-x

https://doi.org/10.1108/LHT-03-2021-0113

https://doi.org/10.1007/s11356-022-19718-6

https://doi.org/10.1007/s11356-022-19628-7

https://doi.org/10.1108/FS-02-2021-0053

https://doi.org/10.1016/j.resourpol.2022.102730

https://doi.org/10.1007/s11356-022-19954-w

https://doi.org/10.1007/s11356-022-20178-1

https://doi.org/10.1007/s11356-022-20922-7

We would like to thank the reviewer for his suggestions. Unfortunately, we did not find the papers closely relevant to our study. However, some of them are very interesting to us as we do research in these areas (environmental sustainability, covid-19) as well.

Methodology: This section is well-written and well-explained. Referring to my comments in the Introduction section, try to explain each abbreviation at first and then use its short form.

Done; see our response above.

Results are explained in detail. Must add much more explanations and interpretations for the results, which are not enough. It is suggested to compare the results of the present research with some similar studies which is done before (more justification is needed).

We extended the interpretation of the results slightly. Unfortunately, no other similar studies are available – outside of the studies we already refer to. We double-checked that fact when updating the literature review using the newest studies, as suggested by Reviewer #2.

Conclusion: Please make sure your conclusions section underscores the scientific value-added of your paper and the applicability of your findings/results, as indicated previously. Please revise your conclusion part into more detail. It would be best if you enhanced your contributions, and limitations, underscore the scientific value-added of your paper, and the applicability of your findings/results and future study in this session.

We rewrote the discussion/conclusion part of the study to communicate the requested components more clearly. Thank you very much for highlighting this issue.

Reviewer #2

References used in the study aren’t recent ones. Please add more of recent research on the topic.

We thoroughly reviewed the recent literature. While no significant new work has been published in the meantime, we still included several recent studies that strengthened our argument.

The motivation to or need to conduct this study is missing.

This comment is clearly related to what Reviewer #1 said as well – we addressed it by considerably rewriting the introduction to clarify why this study is needed.

Please consider correcting grammatical errors as in Line 202

Corrected.

Title of the study doesn’t appear interesting, it should be changed.

We changed the title, and we believe it is much more exciting and attention-drawing than the previous version.

Once again, thank you for your feedback. It considerably helped us in improving our manuscript.

We also formatted the manuscript so it better fits the journal’s template. We want to apologize for omitting a few formatting requirements in our original submission.

Yours sincerely, the authors

---

## [Decision Letter · Decision Letter 1]

20 Jul 2023

PONE-D-22-33605R1Flying high on low cost: Success in low-cost air carriersPLOS ONE

Dear Dr. Jirásek,

Thank you for submitting your manuscript to PLOS ONE. After careful consideration, we feel that it has merit but does not fully meet PLOS ONE’s publication criteria as it currently stands. Therefore, we invite you to submit a revised version of the manuscript that addresses the points raised during the review process.

I recommend that it should be revised taking into account the changes requested by the reviewers. Since the requested changes include valuable and constructive reviews, I would like to give you a chance to revise your manuscript. The revised manuscript will undergo the next round of review by same reviewers.

We look forward to receiving your revised manuscript.

Kind regards,

Baogui Xin, Ph.D.

Academic Editor

PLOS ONE

Reviewers' comments:

Reviewer's Responses to Questions

**Comments to the Author**

1. If the authors have adequately addressed your comments raised in a previous round of review and you feel that this manuscript is now acceptable for publication, you may indicate that here to bypass the “Comments to the Author” section, enter your conflict of interest statement in the “Confidential to Editor” section, and submit your "Accept" recommendation.

Reviewer #1: All comments have been addressed

Reviewer #2: All comments have been addressed

Reviewer #3: (No Response)

Reviewer #4: (No Response)

2. Is the manuscript technically sound, and do the data support the conclusions?

Reviewer #1: Yes

Reviewer #2: Yes

Reviewer #3: Partly

Reviewer #4: Partly

3. Has the statistical analysis been performed appropriately and rigorously? 

Reviewer #1: Yes

Reviewer #2: No

Reviewer #3: Yes

Reviewer #4: Yes

4. Have the authors made all data underlying the findings in their manuscript fully available?

Reviewer #1: Yes

Reviewer #2: No

Reviewer #3: Yes

Reviewer #4: Yes

5. Is the manuscript presented in an intelligible fashion and written in standard English?

Reviewer #1: Yes

Reviewer #2: Yes

Reviewer #3: Yes

Reviewer #4: Yes

6. Review Comments to the Author

Reviewer #1: Authors incorporate all the comments have been addressed . Also , This paper is accepted for publication

Reviewer #2: All comments mentioned in the revision have been addressed in the updated version. I wish the author (s) all the best.

Reviewer #3: Manuscript: PONE-D-22-33605_R1

Title: Flying high on low cost: Success in low-cost air carriers

Overview

The paper employs csQCA (Crisp Set Qualitative Comparative Analysis) to ascertain, from a set of causal indicators determined by the authors, the characteristics indicative of successful low cost carriers (LCCs) in the European airline space. The authors provide a brief description of the steps involved in carrying out a csQCA analysis and then provide detailed results and analysis sections. The authors interpretation of analysis leads to the conclusion that there are two distinct business models leading to success of a LCC.

Concerns

1. The authors adopt the QCA investigative approach and state that they use crisp set QCA. While QCA itself seems a good choice (given the number of cases being investigated) I would like to have seen rationale for the choice of csQCA over fuzzy set fsQCA. The authors point to the use of fsQCA 3.0 software (which support fuzzy sets QCA) in the Methodology section. Are the methods so similar that an analysis tool built for fsQCA can be immediately applied to csQCA formulation?

2. The authors devise 6 conditions and a success attribute which they then use in generating causality rules. Each condition, and the success label, have multiple indicators and calibration criteria. These are used in generating the truth table for each airline. IT is not made clear how the indicators and calibration criteria are combined to determine a binary 0/1 value for the condition for each airline. For instance, is it the case that satisfying a single indicator/calibration criteria is sufficient for the airline to be rated a 1 for the condition?

3. I would like to have seen a discussion on candidate conditions that were considered but rejected, the rationale for such decisions, what POA said about the conditions,...

4. I would like to have seen some analysis of the conditions that showed they are not correlated.

5. The authors make note of the Long Haul condition and the are hesitant to interpret its role in determining success of LCC. Further work needs to be done to include non-European LLCs that do service a wide geographical area - for instance Scoot, Singapore Airline's LCC.

Minor

l-71 between the two spokes -> suggest between any two spokes

l-79 missing full stop at the end of the sentence

l-92 Albers's -> Albers'

l-140 Mill's methods -> need a reference for this

l-214 denoted as "Long-haul" -> should this be denoted as "~Long-haul"

Reviewer #4: This study aims to construct successful LCC business models by identifying distinctive features and limitations using Qualitative Comparative Analysis (QCA). I still have concerns about the novelty and robustness of this study's methodology and confirming the results' reliability. Overall, there are several major issues:

1. Compared to previous research, the contribution of this study needs to be adequately specified. The authors can highlight the significance and uniqueness of this study compared to existing literature, such as:

• Loureiro, S. (2016). In-flight attributes and mindful passengers: Qualitative comparative analysis (QCA) of relationship quality and behavioral intentions configurations. In-flight attributes and mindful passengers: Qualitative comparative analysis (QCA) of relationship quality and behavioral intentions configurations, 1032-1033.

• Hvass, K. A. (2012). A Boolean Approach to Airline Business Model Innovation.

• Ott, U. F., Sinkovics, R. R., & Hoque, S. F. (2018). Advances in qualitative comparative analysis (QCA): Application of fuzzy set in business and management research. C. Cassell, A. Cunliffe, & G. Grandy, The SAGE Handbook of Qualitative Business and Management Research Methods: Methods and Challenges, 414-430.

2. The literature review should examine the relationship between the independent variables utilized in the study and their correlation with the success criteria in Table 2.

3. Please provide a more detailed explanation of the method's development and the software employed in this study. Additionally, clarify the reason for selecting FsQCA 3.0 as the chosen software over alternatives such as nvivo, Atlas.ti, or other similar tools.?

4. In lines 163-168, How can the authors ensure that paid onboard refreshments, the use of secondary airports, tight seat pitches, and seat widths are not determining factors in differentiating between successful and unsuccessful ventures? Are there any references supporting this claim?

5. How was the quantitative value in calibration criteria Table 1 defined or derived? Are these numbers correlated with airline success in previous studies?

6. How were the success calibration criteria in Table 2 defined? For instance, what is the reason behind setting the cost per available seat kilometer to less than 4 U.S. cents? Is there a specific reason for selecting 3.5 points as the calibration value in Customer reviews??

7. Have the authors conducted a sensitivity analysis on the selection of the independent variable calibration (IV) and outcome calibration (DV) to ensure the robustness of the results??

8. The interpretation of the results could be more precise. The authors should explicitly mention the specific result or table to which they are referring in their explanations.

9. Please establish connections between your findings and previous studies. Additionally, provide plausible explanations for the outcomes in the discussion.

7. PLOS authors have the option to publish the peer review history of their article (what does this mean?). If published, this will include your full peer review and any attached files.

Reviewer #1: **Yes: **KASHIF ABBASS

Reviewer #2: No

Reviewer #3: **Yes: **Robert Andrews

Reviewer #4: No

---

## [Author Response · Author response to Decision Letter 1]

16 Aug 2023

The response is contained in the cover letter.

---

## [Decision Letter · Decision Letter 2]

7 Nov 2023

Flying high on low cost: Success in low-cost airlines

PONE-D-22-33605R2

Dear Dr. Jirásek,

We’re pleased to inform you that your manuscript has been judged scientifically suitable for publication and will be formally accepted for publication once it meets all outstanding technical requirements.

Kind regards,

Baogui Xin, Ph.D.

Academic Editor

PLOS ONE

Additional Editor Comments (optional):

Reviewers' comments:

Reviewer's Responses to Questions

**Comments to the Author**

1. If the authors have adequately addressed your comments raised in a previous round of review and you feel that this manuscript is now acceptable for publication, you may indicate that here to bypass the “Comments to the Author” section, enter your conflict of interest statement in the “Confidential to Editor” section, and submit your "Accept" recommendation.

Reviewer #3: All comments have been addressed

2. Is the manuscript technically sound, and do the data support the conclusions?

Reviewer #3: Yes

3. Has the statistical analysis been performed appropriately and rigorously? 

Reviewer #3: Yes

4. Have the authors made all data underlying the findings in their manuscript fully available?

Reviewer #3: Yes

5. Is the manuscript presented in an intelligible fashion and written in standard English?

Reviewer #3: Yes

6. Review Comments to the Author

Reviewer #3: The revised manuscript has addressed all my previous concerns. The paper reads well and is well reasoned. I like the approach and the discussion about conjunction of attributes leading to a conclusion, and the notions of equifinality and asymmetry being used to clarify results.

I note that there are a mix of capitalisation styles in the names of articles in References, as well as one instance [3] where an abbreviated journal title is used. These should be addressed for the final version.

7. PLOS authors have the option to publish the peer review history of their article (what does this mean?). If published, this will include your full peer review and any attached files.

Reviewer #3: **Yes: **Robert Andrews

---

## [Editor Report · Acceptance letter]

13 Dec 2023

PONE-D-22-33605R2 

PLOS ONE

Dear Dr. Jirásek, 

I'm pleased to inform you that your manuscript has been deemed suitable for publication in PLOS ONE. Congratulations! Your manuscript is now being handed over to our production team.

Kind regards, 

on behalf of

Professor Baogui Xin 

Academic Editor

PLOS ONE